# Genetic Update and Treatment for Dystonia

**DOI:** 10.3390/ijms25073571

**Published:** 2024-03-22

**Authors:** Jan Koptielow, Emilia Szyłak, Olga Szewczyk-Roszczenko, Piotr Roszczenko, Jan Kochanowicz, Alina Kułakowska, Monika Chorąży

**Affiliations:** 1Department of Neurology, Medical University of Bialystok, 15-276 Bialystok, Poland; emiliaszylak96@gmail.com (E.S.); jan.kochanowicz@umb.edu.pl (J.K.); alina.kulakowska@umb.edu.pl (A.K.); monika.chorazy@umb.edu.pl (M.C.); 2Department of Synthesis and Technology of Drugs, Medical University of Bialystok, Kilinskiego 1, 15-089 Bialystok, Poland; olga.szewczyk@sd.umb.edu.pl (O.S.-R.); piotr.roszczenko@sd.umb.edu.pl (P.R.)

**Keywords:** focal dystonia, blepharospasm, cervical dystonia, oromandibular dystonia, spasmodic dysphonia

## Abstract

A neurological condition called dystonia results in abnormal, uncontrollable postures or movements because of sporadic or continuous muscular spasms. Several varieties of dystonia can impact people of all ages, leading to severe impairment and a decreased standard of living. The discovery of genes causing variations of single or mixed dystonia has improved our understanding of the disease’s etiology. Genetic dystonias are linked to several genes, including pathogenic variations of VPS16, TOR1A, THAP1, GNAL, and ANO3. Diagnosis of dystonia is primarily based on clinical symptoms, which can be challenging due to overlapping symptoms with other neurological conditions, such as Parkinson’s disease. This review aims to summarize recent advances in the genetic origins and management of focal dystonia.

## 1. Introduction

Dystonias are a group of disorders characterized by abnormal, repetitive movements or postures caused by excessive muscle contractions [1]. Dystonia could be unrelated to other diseases, with initial symptoms usually appearing in childhood, but also may be caused by nervous system damage and have no genetic basis. In addition to underlying conditions and diseases such as brain tumors, cerebral palsy, Parkinson’s disease, stroke, multiple sclerosis, hypoparathyroidism, or vascular malformations, other factors that can contribute to brain conditions include certain medications, exposure to carbon monoxide, cyanide, manganese, or methanol [2].

Focal dystonias are the most common forms of dystonia and typically affect a single body region, such as the neck (cervical dystonia), the upper part of the face (blepharospasm), the larynx (spasmodic dysphonia), or the top of a limb (focal hand dystonia) (see Figure 1). The pathophysiology of focal dystonias is hypothesized to involve the brain’s motor system [3,4].

Segmental dystonia affects two or more regions of the body that are contiguous or in proximity. In up to 30% of cases, areas near the primary site of focal dystonia can also experience spasms. A common form of segmental dystonia affects the lower face, jaw, lips, and eyelids. Other forms of dystonia include hemidystonia, which affects half of the body; multifocal dystonia, which affects two or more distant body parts; and generalized dystonia, which progresses to other body parts [5].

Cervical dystonia (CD) is the most common type of focal dystonia. It is characterized by involuntary muscular contractions that cause aberrant head, neck, and shoulder movements and postures, as well as tremors and pain [6]. Pain is a significant cause of disability and social isolation in patients with cervical dystonia and is often the primary reason for seeking treatment. Pain is often cited as a challenging and debilitating aspect of dystonia symptoms in patient surveys, with a significant impact on daily activities and work. The mechanisms of pain in cervical dystonia (CD) may be muscle-based or non-muscle-based. There is accumulating evidence that non-muscle-based mechanisms, such as dysfunction of descending pain inhibitory pathways, as well as structural and network changes in the basal ganglia, cortex, and other areas, may contribute to pain in CD alongside prolonged muscle contraction [7].

Blepharospasm (BSP) is an adult-onset focal dystonia characterized by involuntary eyelid closing and orbicularis oculi muscle spasms [8]. In 1910, Henri Meige described ten patients with blepharospasm who experienced involuntary eyelid closing along with jaw muscular contractions. Meige referred to this condition as ‘Convulsions de la Face’ in his study [9]. The most prominent symptom is spasms of ocular closure. Another symptom of this condition is frequent blinking, which may occur independently or in conjunction with spasms. Additionally, individuals may experience difficulty opening their eyes without the presence of orbicularis oculi spasm [10].

Oromandibular dystonia (OMD) is a rare type of focal idiopathic dystonia that was first clinically discovered at the turn of the twentieth century. However, some features of OMD, such as the high risk of oral trauma associated with the development of motor symptoms [11], remain unexplained. OMD is a type of focal dystonia that causes repeated and continuous spasms of the facial, lingual, and masticatory muscles. This results in uncontrolled movements of the jaw, including opening and closing, retractions, protrusions, or a combination of these. OMD can occur alone or in conjunction with limb or trunk dystonia, which may be caused by toxins, neurodegenerative, metabolic, infectious, postanoxic, post-traumatic stress disorder, stroke, status epilepticus, primary faciobrachial, or generalized dystonia. In some patients, the cause of OMD cannot be determined [12].

Spasmodic dysphonia (SD) is a type of focal dystonia that causes excessive or incorrect contraction of laryngeal muscles during speech. SD is characterized by either inadequate glottic closure (adductor type) or abrupt opening of the vocal folds (abductor type). The main symptom of adductor-type SD is a strained or strangled voice, while a breathy or absent voice characterizes abductor-type SD. Ninety-seven percent of all patients with SD have Adductor-type SD, with 70% of them showing abnormal contractions of extra laryngeal muscles [13].

Focal Hand Dystonia (FHD) is a movement disorder characterized by involuntary movements, cramps, and spasms. It is linked to pathological neuronal microcircuits in the cortical somatosensory system. Researchers can explore individual cortical layer and column neuronal microcircuits using invasive preclinical techniques [14]. The most common type of focal, task-specific dystonia is writer’s cramp. Clinical manifestations of focal dystonia include a range of distinct upper extremity dystonic patterns, such as difficulty writing or holding a writing tool. Symptoms often develop in the context of repeated hand movements and increased writing demands. Musicians are a group that is particularly susceptible to this condition, which can manifest when highly skilled motions are performed. The pathophysiological factors that lead to focal dystonia in musicians are currently unclear. However, recent neurophysiological research and functional neuroimaging have revealed increasing evidence of abnormalities in sensory-motor unity, cortical and subcortical inhibition processes, and the handling of sensory information in this disease [15]. Although focal task-specific dystonia of the hand in musicians, known as musicians’ hand dystonia, has been well studied, musicians may also experience issues with their laryngeal muscles [16], lower limbs [17], and embouchure [18]. Embouchure dystonia is a condition where musicians experience dystonia in their perioral and facial muscles while playing embouchure instruments. It is crucial to diagnose this condition promptly because if the musician continues playing the instrument, the dystonia could become non-specific and persistent. Drummers are the only musicians documented to have task-specific dystonia of the lower limbs. The diagnosis is based on the electromyogram of the affected muscles during the task. The term ‘singer’s dystonia’ refers to a type of laryngeal dystonia that is specific to singing tasks [19]. Task-specific dystonia may be noticeable during fine motor control tasks such as writing, playing an instrument, and other chores. It can also occur during specific sports and may even be detrimental to the careers of professional athletes. Therefore, neurologists who specialize in movement disorders and sports physicians must have a thorough understanding of the symptoms and presentation of dystonia related to sports [20].

In cases of isolated focal dystonia, the site of onset is highly associated with the likelihood of the dystonia spreading and affecting subsequent body parts. Patients with BSP had more than twice the chance of dystonia spreading compared to those with cervical, hand, or laryngeal dystonia at onset. Additionally, findings indicate that individuals with BSP tend to spread symptoms to the neck and oromandibular area, while patients with CD tend to spread to the hand. Patients with laryngeal dystonia or hand, alternatively, tend to spread to the neck. Moreover, our results support the idea that a higher risk of spread may be associated with alcohol responsiveness and a positive family history [21]. There is a risk of misdiagnosis of focal dystonia. An analysis of 2916 cases diagnosed as focal dystonia found that 28.3% of those diagnosed with CD had dystonia outside the neck region. This observation highlights the need for more specific guidelines to define this disorder [22].

A small proportion of individuals diagnosed with idiopathic adult-onset dystonia that affects two body parts experience the spread of symptoms to a third body site [23]. Patients with focal onset dystonia appear to have a higher risk of spread to a third body location, regardless of the site of dystonia onset and other demographic/clinical characteristics, in contrast to those with segmental/multifocal dystonia at onset [24].

## 2. Classification and Comorbidities in Dystonia

The causes of dystonia are diverse. Albanese et al. classify the causes into two subcategories: inherited vs. acquired disease, and the presence or absence of degenerative or structural lesions. The use of the term ‘primary dystonia’ has become less common since the release of the more modern Albanese classification, which provides a different classification that might only differentiate between acquired and hereditary causes of dystonia [25,26].

Dystonia can be acquired from any process that damages the basal ganglia. These processes include vascular causes, such as perinatal hypoxic-ischaemic encephalopathy (which is the primary cause of dystonic cerebral palsy) and stroke (which is also a significant cause of dystonia in all age groups); toxic causes, such as heavy metal poisoning; iatrogenic causes, such as the use of antipsychotic drugs; traumatic brain injury; or the presence of a space-occupying lesion [27]. Concepts of the neuroanatomical basis of dystonia have changed from a relatively limited focus on basal ganglia dysfunction to a more comprehensive motor network model in which the basal ganglia, cerebellum, cerebral cortex, and other brain regions play important roles. Our knowledge of the underlying physiological abnormalities has increased and several changes in brain signaling in the cortex, cerebellum, and basal ganglia have been identified [28].

Genetic dystonias encompass both situations where the primary symptom is dystonia (e.g., *TOR1A*, *ANO3* mutations, etc.) and conditions where dystonia is a co-occurring feature of a more complex condition. There is some overlap in the neurological, neuropsychiatric, and systemic characteristics associated with almost all hereditary dystonias, making this distinction imprecise. Dystonia is one of many features of a wide range of genetic problems, such as neurotransmitter abnormalities, neurometabolic disorders, and neurodegenerative diseases. While many diseases have early onset, others, like Huntington’s disease (HD), may not exhibit symptoms until late adulthood [29].

Parkinson’s disease and dystonia are inextricably related [30]. There is strong evidence for the importance of dopaminergic dysfunction in the pathogenesis of dystonia, suggesting a possible pathophysiological overlap with parkinsonism. This evidence comes from animal models, neuropathological, neurophysiological, neuroimaging, and clinical studies. Clinically, parkinsonian features are an important component of some combination dystonias, including dopa-responsive dystonia. Both inherited isolated dystonias and idiopathic dystonias may have mild Parkinsonian features. Idiopathic isolated dystonia also often causes postural, action, and resting tremors. In addition to being common in idiopathic isolated dystonia, postural, action, and rest tremor may mimic PD tremors and be the cause of “scans without evidence of dopaminergic deficit”. The clinical spectrum of motor features in dystonia will be broadened by the identification and better clinical characterization of Parkinsonian features in idiopathic and hereditary isolated dystonia. This may prevent misdiagnosis and guide future treatment research [31,32].

Both dystonia and LID (levodopa-induced dyskinesia) are hyperkinetic movement disorders. LID relates to Parkinson’s disease (PD), arising due to chronic levodopa medication, and can be dystonic or choreiform. LID and dystonia share several phenomenological characteristics and processes. LID and dystonia are caused by an integrated circuit that includes the cortex, basal ganglia, thalamus, and cerebellum [33].

## 3. The Genetic Background of Dystonia

Pathogenic variations in multiple genes can cause isolated dystonia, and the number of identified dystonia genes is expanding [4], but the prevalence of recognized monogenic dystonias is limited. Notably, up to 30% of individuals with idiopathic dystonia report having first- or second-degree family members with dystonia [34,35,36], implying an inherited/genetic role in these cases as well. Furthermore, the genetics of dystonia are characterized by low penetrance, possibly leading to an underestimation of hereditary causes [37]. Pathogenic variants in dystonia genes have been linked to a variety of pathways, including gene transcription during neurodevelopment, endoplasmic reticulum stress response, calcium homeostasis, striatal dopamine signaling, and autophagy [38]. Currently, genes *THAP1*, *GNAL*, *TOR1A*, and *ANO3* are linked to isolated dystonia and have undergone extensive validation. *KMT2B* and *VPS16* are also well validated, bringing the total number of monogenic dystonias to at least six, plus a longer list of combined dystonias such as *PANK2*, *ATP1A3*, *SCGE*, etc. Monogenic sporadic and inherited types of isolated dystonia are caused by rare variations in these genes; frequent variations may impart risk and suggest that dystonia is a polygenic feature in a subset of instances [39].

(a)TOR1A (torsin 1A—related to endoplasmic reticulum stress response)TOR1A, the initially identified dystonia gene, was discovered about 25 years ago on human chromosome 9q34. A distinct 3-bp deletion results in the loss of one of a pair of glutamic-acid residues in the conserved C-terminus of torsinA. Generalized dystonia affecting the lower limbs that begins in childhood is linked to the GAG deletion in TOR1A [40]. A meta-analysis has shown that the TOR1A variants rs1182 and rs1801968 have a significant influence on the development of the writer’s cramp and focal dystonia, respectively. A functional missense variant known as rs1801968 exists in exon 4 of the *TOR1A* gene, where histidine is substituted by aspartic acid at position 216. On the other hand, the precise functional effect of rs1182 on primary dystonia is still unknown, as there is no known functional effect of rs1182 on TOR1A expression and function [41].(b)THAP1 (THAP domain-containing protein 1—participates in neurodevelopment)Most known mutations in the thanatos-associated protein 1 (THAP1) gene are missense or out-of-frame deletions, while intronic or splice site alterations are rare [42]. Many *THAP1* variants, including truncating mutations, have been classified as possibly harmful since they have only been reported in individuals with no information on segregation. Although truncating mutations normally cause a loss of function, they are not always associated with disease. Mutations of the *THAP1* gene are characterized by juvenile-onset global dystonia, and the condition is mostly linked to missense variants in the gene [43].(c)KMT2B (lysine-specific methyltransferase 2B—participates in neurodevelopment)KMT2B-related dystonia is a movement condition that typically begins in childhood. It is characterized by a progressive course, often starting as focal dystonia in the lower limbs and advancing to generalized dystonia with significant involvement of the cervical, cranial, and laryngeal regions. A proband with either a heterozygous pathogenic variation in KMT2B or a heterozygous interstitial deletion of 19q13.12, which includes a *KMTB2* whole-gene deletion, is a candidate for diagnosis of this disease. [44].(d)GNAL (guanine nucleotide-binding protein alpha-activating activity polypeptide—involved in striatal dopamine signaling)Previous reports suggest that GNAL variations have been detected in less than 2% of dystonia cases of European origin and contribute to disease in late adulthood [45]. However, in the Hungarian cohort study, it was found in a higher prevalence than previously reported. All three mutations identified there (c.677G>T (p.Cys226Phe), c.1315G>A (p.Val439Met) and c.1288G>A (p.Ala430Thr)) are thought to be pathogenic, and the clinical profile fits the characteristics already linked with GNAL variants [46].(e)ANO3 (anoctamin—engaged in calcium homeostasis)*ANO3*-linked dystonia has an autosomal-dominant inheritance pattern, yet only heterozygous variations have been identified. The mutation is connected with a very broad phenotypic spectrum with a severe childhood-onset disease (often due to de novo variants) and late-onset focal forms, often with tremors. Most individuals experienced multifocal or segmental dystonia, with focal or generalized distribution being less prevalent. The most common onset was cervical dystonia, followed by upper limb dystonia. Lower limb onset has also been recorded in several cases and appears to relate to a younger age (under 20) [43].(f)VPS16 (PS16 core subunit of corvet and homotypic fusion and vacuole protein sorting (HOPS) complexes—participates in autophagy)*VPS16* variations were discovered in the context of early-onset generalized dystonia accompanied by lysosomal dysfunction [47]. Nevertheless, *VPS16* variations have recently been discovered among patients suffering from focal dystonia [48]. Furthermore, these results imply that *VPS16* variations should be evaluated in cases of focal dystonia. It is also noteworthy that a positive family history has been linked to the spread of dystonia from one body location to another [21].(g)TSPOAP1 (TSPO Associated Protein 1- involved in the regulation of presynaptic calcium ion concentration)Homozygous frameshift, nonsense, and missense mutations in *TSPOAP1*, the gene that encodes the active-zone Rab3 interacting molecule-binding protein 1 (RIMBP1) are connected to autosomal recessive dystonia. Individuals with loss-of-function mutations displayed progressive generalized dystonia with juvenile-onset, along with cerebellar atrophy and intellectual impairment. On the other hand, individuals with p.Gly1808Ser, a pathogenic missense mutation, displayed isolated focal dystonia that developed in adulthood. Complete deletion of RIMBP1 impedes neurotransmission via the link between presynaptic action potentials and the exocytosis of synaptic vesicles, as well as via presynaptic voltage-gated Ca^2+^ channels (VGCCs). This results in lower numbers of cerebellar synapses, decreased Purkinje cell dendritic arborization, and motor disorders resembling dystonia in mice [49].(h)ATP1A3 (Na(+)/K(+)-ATPase alpha3 catalytic subunit- maintaining the Na and K ion electrochemical gradients throughout the plasma membrane)In addition to the originally identified phenotypes of rapid-onset dystonia-parkinsonism, alternating hemiplegia of childhood, cerebellar ataxia, areflexia, pes cavus, optic atrophy, and sensorineural hearing loss syndrome, ATP1A3 is linked to a wide range of primarily neurologic disorders. It is difficult to determine the pathogenicity of an ATP1A3 variation discovered in an undiagnosed patient due to this phenotypic heterogeneity. Exons 8, 14, 17, and 18 are the specific rapid-onset dystonia parkinsonism-related mutation hotspots for ATP1A3, with T613M and I578S being the most common variations [50]. A813V (2438C>T), a novel mutation in ATP1A3, was discovered. In the ATP1A3 A813V mutation, the initial alanine was changed to valine. Crystallography showed that the side chain of valine was significantly larger than that of alanine, which changed the shape of the ATP1A3 protein and reduced its biological activity [51].(i)GCH1 (guanosine triphosphate cyclohydrolase I gene—biosynthesis of neurotransmitters)The GCH1 gene codes for the expression of the enzyme GTP cyclohydrolase 1. This enzyme is involved in the first of three steps in the formation of a compound known as tetrahydrobiopterin (BH4). For the synthesis of important neurotransmitters like dopamine, serotonin, and nitric oxide synthases, BH4 is a necessary component. Neuropsychiatric disorders such as depression, hyperalaninemia, Parkinson’s disease, and dopa-responsive dystonia are caused by a lack of BH4, whose synthesis is hampered by *GCH1* deficiency [52].(j)SGCE (Epsilon-sarcoglycan—stability of striated muscle)A movement condition called myoclonus-dystonia syndrome is linked to abnormalities in the SGCE gene. The gene produces the transmembrane protein ε-sarcoglycan, which has an isoform unique to the brain. Because of maternal imprinting, this genetic abnormality has reduced penetrance and is transmitted autosomally and dominantly [53]. Leg involvement is less likely in the myoclonic jerks typical of SGCE myoclonus-dystonia (SGCE-M-D), which typically impact the neck, trunk, and upper limbs. The writer’s cramp and/or cervical dystonia are symptoms of further focal or segmental dystonia in about half of the affected patients [54].The hereditary basis of dystonia is not yet fully understood. A 2020 exome-wide sequencing study provides support for this view. This study has established a link between dystonia and 11 genes associated with neurodevelopmental disorders. These genes include AUTS2, CHD8, and ZEB2, whose associated trait manifestations have been more thoroughly characterized, as well as DHCR24, MORC2, GRID2, MSL3, PAK1, TECPR2, PPP2R5D, and ZMYND11, for which only a small number of families with pathogenic variants have been reported. Out of the 13 cases with mutations in these genes, 12 presented combined dystonia, and all of them had comorbidities that were not related to movement disorders [55].

## 4. Diagnostic of Dystonia

Finding an underlying etiology for focal dystonia requires accurate phenotyping of the condition, but specific tests are also needed due to the heterogeneity. The last several years have seen a rise in interest in both motor symptoms and the related nonmotor symptoms, as well as their negative effects on quality of life. The growing number of recently identified genes linked to dystonia complicates the diagnosis process. The goal of recent initiatives has been to advance the development of algorithms and recommendations to help with diagnosis and with utilizing diagnostic instruments [56].

The diagnostic and therapeutic recommendations algorithm by the Italian Society of Neurology, the Italian Academy for the Study of Parkinson’s Disease and Movement Disorders, and the Italian Network on Botulinum Toxin suggested division on two axes: clinical features and etiology. In the first axis, a description of the body distribution provides an assessment of how each patient’s motor symptoms develop over time. The following forms are identified in the present classification: focal, segmental, multifocal, generalized, and hemidystonia. The second axis, which addresses the etiology of dystonic disorders, is constantly changing in light of new scientific knowledge [57] (Figure 2).

While imaging and laboratory tests are usually normal in patients with dystonia, there is no good diagnostic test for idiopathic dystonia. Voluntariness, mirror dystonia (unilateral abnormal posture induced by contralateral movements), overflow dystonia (involuntary contraction of muscles anatomically distinct from the primary movement), and, in certain cases, dystonic tremor (movements are oscillatory but not strictly rhythmic, jerky and patterned) are key clinical features of dystonia. In addition, there may be a “null point” where a person’s movements are worse in one position and better in another due to dystonia or dystonic tremor. The presence of relaxation techniques, also known as sensory tricks is another essential component. It is important to ask precise questions and evaluate sensory tricks through investigation. Clinical factors (onset age, body distribution, temporal pattern, and related clinical aspects) and origin (such as nervous system pathology or whether the disorder is inherited, acquired, or idiopathic) are used to categorize dystonia [55]. Dystonia diagnosis is a difficult task. Patients with functional dystonia (also known as “psychogenic”), scoliosis, myoclonus, tics, Parkinson’s disease, and headaches are frequently misdiagnosed. An important point of differentiation is fixed dystonia. This was defined as an immobile dystonic posture that did not return to a neutral position at rest [58,59] (Figure 3).

Four different forms of dystonia can be distinguished based on their temporal pattern. The occurrence and intensity of dystonia follow a consistent pattern that is similar throughout the day. Dystonia that is specific to an action or task, such as typing, singing, performing an instrument, or writing, only manifests itself during that specific activity. Paroxysmal dystonia is characterized by abrupt, distinct dystonia episodes that end with a return to the neurologic baseline. In dopa-responsive dystonia, diurnal variation—mild symptoms upon arising that get worse as the day goes on—is typically observed. One can treat dystonia alone or in conjunction with another movement problem. Apart from tremor, which is typically phenomenologically dystonic in origin, dystonia is the sole clinical motor symptom associated with solitary dystonia. However, various movement disorders such as myoclonus, parkinsonism, ataxia, or chorea/dyskinesias are linked to combined dystonia. While neuropsychiatric symptoms are often linked to dystonia, cognitive impairment is often limited to degenerative/progressive dystonias. Differentiating dystonia can be based on how quickly symptoms manifest. At least initially, focal dystonias usually occur after a gradual or subacute deterioration of symptoms, which is then followed by a plateau. Although it usually plateaus, further expansion to segmental/multifocal dispersion is possible [62].

Not only can Parkinson’s disease manifest with concurrent dystonia, but in some circumstances, parkinsonian characteristics might be mistaken for dystonia. Dystonia is a commonly present feature in patients with Parkinson’s disease, which can complicate diagnosis. Of these, 30% generate off-state dystonia (which occurs when there are low levels of plasma levodopa, and respond well to levodopa treatment), and 10% to 15% may develop (usually lower) limb dystonia. Other possible complications include blepharospasm, apraxia of eyelid opening (primarily in atypical parkinsonism), as well as truncal/axial dystonia, which includes camptocormia and Pisa syndrome [63,64].

## 5. Treatment for Dystonia

BoNT (botulinum neurotoxin) has been utilized since the 1980s and remains the primary treatment for focal dystonia. The mechanism of action of BoNT involves extracellular binding to glycoprotein structures on cholinergic nerve terminals, cleavage of components of the SNAP (soluble N-ethylmaleimide-sensitive factor attachment protein) receptor complex, and neuromuscular transmission by intracellular blockade of acetylcholine release. The main impact of BoNT therapy is the reduction of aberrant activity in the precuneus, the sensorimotor cortical and subcortical regions, and the prefrontal and cerebellar regions [16,65]. Its efficacy has been demonstrated numerous times in multiple research studies and trials, including long-term follow-ups also demonstrating continuing safety and efficacy after as much as two decades of therapy [66,67,68,69]. BoNT formulations are currently accessible in a variety of forms (Figure 4).

Nonetheless, a sizable fraction of patients receiving BoNT therapy fail to respond to treatment either initially or later [71]. There could be a variety of causes for this, some of which are yet unknown [72]. BoNT therapy is difficult; choosing the right muscles to inject and administering the neurotoxin precisely are essential steps toward a successful course of treatment. Certain dystonic symptoms, such as tremulous dystonia or complex multiaxial dystonic movements, are not well suited for treatment with selective chemodenervation. A further consideration is the issue of probable antibody development during long-term BoNT therapy because of its immunogenicity [73]. However, the actual role of immunogenicity in the overall issue of treatment refractoriness can only be evaluated to a limited level, since a recent meta-analysis suggests that the predicted detection frequency of neutralization of BoNT in dystonic disorders is only approximately 1% [74]. Additionally, not all aspects of anti-BoNT antibody formation are fully understood. However, some factors that are thought to play a role include those related to applications, like brief injection cycles or large doses per injection, while additional variables have to do with the toxin itself, like its composition, production, and storage [75].

According to established dystonia treatment guidelines, deep brain stimulation (DBS) can be used as an alternative option if BoNT treatment fails [76,77]. BoNT therapy focuses on the effector organ muscle and is merely symptomatic, whereas DBS acts on the brain to correct the network dysfunction thought to be the cause of dystonic movement disorders [78]. DBS has been demonstrated to enhance symptom management in focal [79], segmental [80], and generalized [81] dystonia. Research studies were able to demonstrate a statistically significant improvement in dystonic symptoms with DBS. Data on safety and tolerability seemed sufficient [82]. Hereditary or idiopathic segmental or generalized dystonia has been treated with bilateral GPi DBS. Patients have shown a significant improvement in motor function, disability, and activities of daily living in both hereditary and idiopathic dystonia. Overall, the statistics suggest a 65% reduction in symptoms, which on average is long-lasting [83]. The follow-up was performed for 14 patients who received stimulation in GPi, STN, or both and had cervical or generalized dystonia. The average duration of subsequent studies was almost ten years. Results confirm that STN-DBS and GPi-DBS have comparable long-term effects and are safe for up to 15 years in the treatment of dystonia. It has been demonstrated that STN is a feasible, safe, and effective target that can be utilized in place of GPi in the treatment of both generalized dystonia and adult-onset cervical dystonia [84].

As existing drugs are not suitable for many patients or become ineffective over time, attempts are being made to administer medicines previously developed for non-dystonia purposes, such as serotonin receptor agonists, various GABAergic pharmaceuticals (e.g., zolpidem or sodium oxybate), and glutamate regulating treatments, could be viable alternatives to traditional dystonia pharmacotherapeutics [85].

In 1969, Siegfried et al. reported Yo thalamotomy for writer’s cramp and observed a significant improvement in patients. The effectiveness of Vothalamotomy for treating FHD in musicians, hair stylists, watchmakers, and table tennis players has now been demonstrated in several publications. In patients with aberrant posterior vo neuronal activity, lesioning of the posterior vo resulted in an instantaneous reduction of dystonia symptoms [86]. MR-guided focused ultrasound (MRgFUS) thalamotomy, which allows intracranial focal lesioning without an incision, is a less invasive and successful approach to treating tremors. Similar to radiofrequency thalamotomy, MRgFUS thalamotomy causes thermal lesions and is expected to have effects on FHD comparable to those of radiofrequency Vo-thalamotomy [87].

It has been suggested that targeting the metabotropic glutamate receptor type 5, or mGlu5, may be a viable therapeutic strategy for the treatment of several neurodegenerative diseases. Novel allosteric modulators have been shown to reduce disease-related pathology and improve cognitive function in preclinical models [88]. Dipraglurant, which affects the mGluR5 by negative allosteric modulation, contributes to the development of dystonia in DYT1 rat models, and sodium oxybate has been examined in patients with alcohol-responsive spasmodic dysphonia with promising results [89]. A summary of the indications for each drug is included in Table 1 (where “++” stands for high level of benefit, + means mild to moderate benefits for use and “–“stands for no current evidence to support use).

## 6. Personal Approach to Dystonia Therapy

A growing number of healthcare providers are using patient journey maps as a tool to help them tailor their services to the needs of their patients. Five important phases of the patient experience were validated by the qualitative analysis of the patient survey: the onset of symptoms, the diagnosis and therapeutic relationship with medical experts, the beginning of CD treatment, the initiation of care for CD, and living with treated CD. After the commencement of symptoms, the survey participants recounted visiting their family physician several times, who prescribed strong analgesics and muscle relaxants and referred them to as many as ten different specialists for a diagnosis. Greater than half (53.3%) of participants reported at least one misdiagnosis. While respondents expressed pleasure at receiving a diagnosis, many expressed confusion about the prognosis and available treatments; 46.7% of respondents felt their neurologist did not devote enough time to answering their questions. Physiotherapy, psychotherapy, and other complementary techniques were recommended by certain neurologists, but botulinum toxin was always mentioned as the primary therapeutic choice. But frequently, patients were left to look for supplemental services on their own. With BoNT treatment, patients experienced a “rollercoaster” of relief; but, by the conclusion of an injection cycle, symptoms (and the ensuing impact on daily living) would return [91].

A multidisciplinary approach involving occupational therapy, speech therapy, physical therapy, behavioral therapy, and psychology procedures is necessary in complex cases [92]. According to available data, rehabilitation may be useful in restoring voluntary motor control and proper posture as well as in minimizing dystonic movements (particularly when used in conjunction with BoNT). Increased pain management and averting further osteoarticular problems have been shown to have benefits of rehabilitation. According to an American Dystonia Society survey, symptoms have improved in 62% of patients with upper limb dystonia and 74% of patients with CD who were referred for physiotherapy treatment. According to a few well-conducted studies, physiotherapy may boost the benefits while potentially lowering the dosage and frequency of treatment. Although a consensus in this sector is currently missing, certain particular rehabilitation treatments for various kinds of focal dystonia have been described. A customized approach should be used while providing outpatient rehabilitation therapy (ideally in the patient’s community). The severity of each instance should determine how often and how intensely the treatment is administered. While some individuals may only require rehabilitative treatment for a few weeks, others may require it for months or years [93,94,95].

There are no reports of attempts being made to introduce gene therapies for diagnosed mutations in identified genes. The use of nanoparticles or adenoviruses as a vector with appropriate genetic material could be a promising therapy in dystonia caused by amendments in genes. Currently, the most developed field of gene therapies is attempting to introduce gene therapy using adenoviruses and nanoparticles in an anticancer approach [96]. The use of adenoviruses as carriers is one of the delivery strategies; this method has been successfully applied in a formulation known as gendicine. Gentacine is used to treat patients with head and neck squamous cell carcinoma associated with TP53 gene mutations. It is produced by Shenzhen SiBiono GeneTech and consists of recombinant adenovirus (rAd-p53) modified to encode the p53 protein. [97]. Even while adenoviruses have shown promising results in preclinical research, they are not always appropriate carriers. As a result, gene therapy may be constrained in the later phases of human trials if an efficient delivery method is not found [98]. One way to distribute the genes more efficiently would be to use nanoparticles (NPs), which show better absorption by cells and increase the stability of the provided particles. For a gene delivery system to work, several conditions must be met. The vector should not be toxic or immunogenic, allowing for several injections if needed. The gene vector should ideally be delivered systemically and precisely targeted [99].

## 7. Conclusions and Future Perspectives

The mechanism behind dystonia is becoming clearer as genetic types of the disease are discovered. The protein products of many genes may explain the underlying mechanism of dystonia and provide a way to permanently relieve patients’ symptoms. However, not all mutations and genes involved in the development of focal dystonias are known, which means that further research into the genomic background is needed. Symptomatic treatments for dystonia include drug therapy with anticholinergics, intramuscular botulinum toxin injection, and DBS. However, there is a significant therapeutic gap in current dystonia treatment options, and reliable biomarkers and better treatments are needed. As most people with focal dystonia are treated with BoNT, testing new medications is challenging.

## Figures and Tables

**Figure 1 ijms-25-03571-f001:**
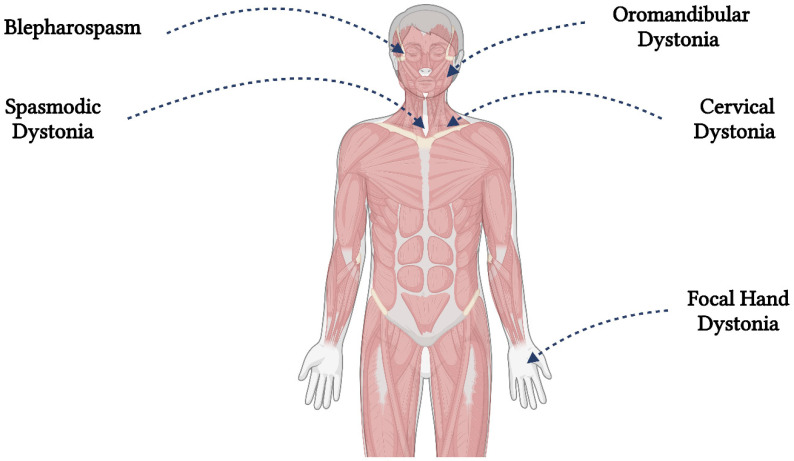
The most frequent locations of dystonia (Created in BioRender.com).

**Figure 2 ijms-25-03571-f002:**
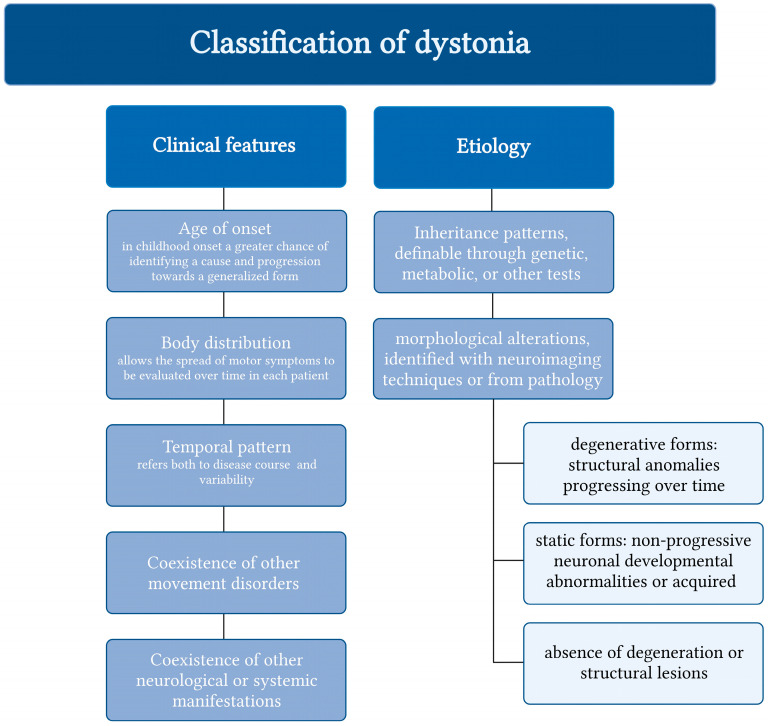
Diagnostic and therapeutic recommendations algorithm by the Italian Society of Neurology, the Italian Academy for the Study of Parkinson’s Disease and Movement Disorders, and the Italian Network on Botulinum Toxin (Created in BioRender.com).

**Figure 3 ijms-25-03571-f003:**
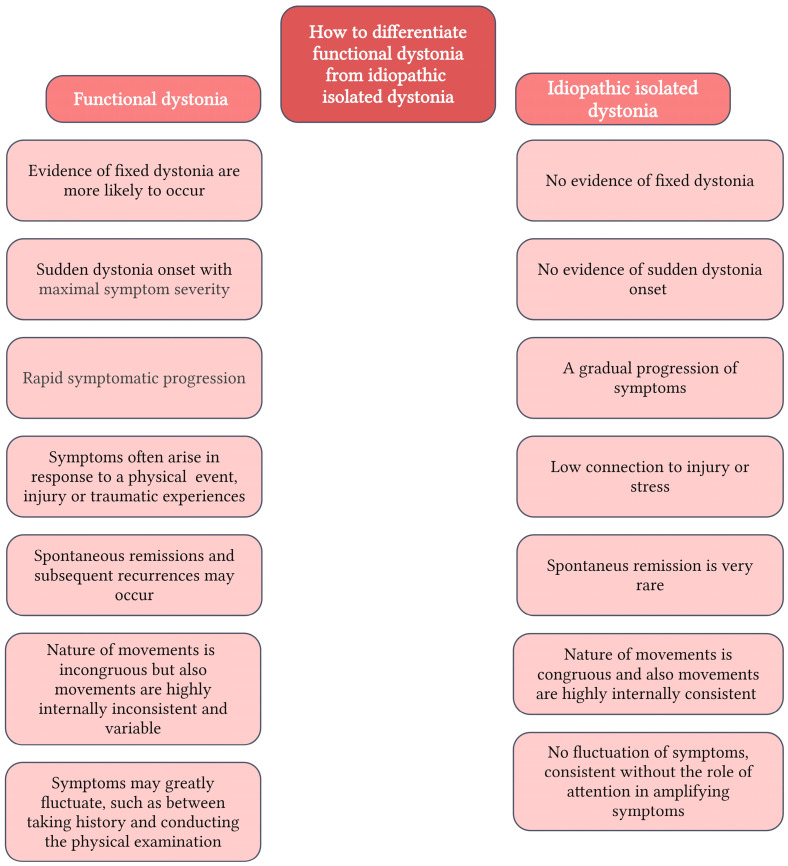
How to differentiate functional dystonia from idiopathic isolated dystonia [60,61] (Created in BioRender.com).

**Figure 4 ijms-25-03571-f004:**
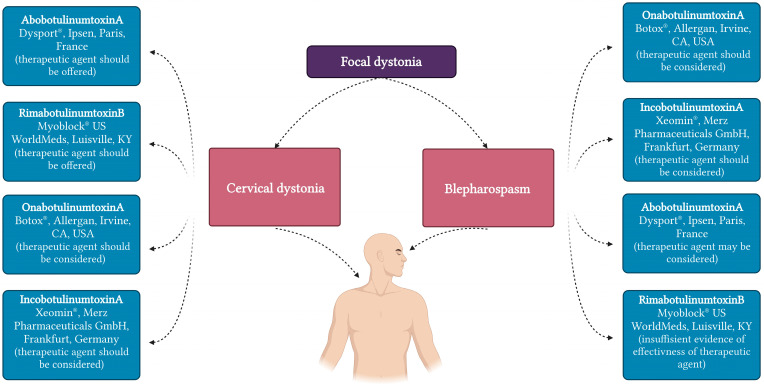
Report of the Guideline Development Subcommittee of the American Academy of Neurology [70]. Recommendations for the use of the listed formulations are given in parentheses (Created in BioRender.com).

**Table 1 ijms-25-03571-t001:** Summary of potential benefits of treatment [90] based on the last guidelines for dystonia diagnosis and therapy [76].

Treatment	Cervical Dystonia	Oromandibular Dystonia	Blepharospasm	Spasmodic Dysphonia	Focal Hand Dystonia
BoNT	++	++	++	++	++
DBS	++	+	+	–	++
Trihexyphenidyl	++	–	–	–	++
Tetrabenazine	++	–	–	–	–
Clonazepam	++	–	++	–	–
Baclofen	–	++	–	–	++
Levodopa	–	–	–	–	–
Amantadine	+	–	–	–	–
Haloperidol	–	–	–	–	–

## Data Availability

Not applicable.

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
