# Peer review of "Genetic Update and Treatment for Dystonia"

_ijms, 2024, doi:10.3390/ijms25073571_

Round 1

Reviewer 1 Report

Comments and Suggestions for Authors

Issues:

1)    My biggest criticism of this work is that the title indicates that the manuscript will focus on focal dystonias, while the main text of this manuscript feels more focused on dystonia in general. I think the authors should focus more on focal dystonias or change their title accordingly. 

2)    Line 13: The term “Foundation genetic dystonias” is unclear to me. What do the authors mean?

3)    Lines 14-15: When the authors state, “which can be challenging”, do they mean the “Diagnosis” or the “clinical signs”?

4)    Line 62: The authors should replace “Oromandibular dystonia (OMD)” with “OMD”, as they already defined “OMD” at the start of this paragraph.

5)    Line 66: When the authors use “toxic”, do they mean “toxins”?

6)    Line 67: When the authors use “post-traumatic”, do they mean “post-traumatic stress disorder” or something else? 

7)    Lines 109-111 and 118-119: The authors use of the term “spread” is confusing in these places. Are they meaning to say that the symptoms are spreading?

8)    Lines 110-111: The authors might consider replacing “on the other hand” here with “alternatively”, as it reads a bit confusing considering the use of “hand” just prior.

9)    Line 125: The authors should replace “control” with “controls” here to maintain subject-verb agreement. In addition, it seems problematic that the authors limit their discussion to the basal ganglia, as several other brain regions (not to mention the spinal cord) are known to be involved in dystonia pathogenesis. The authors should incorporate a more holistic view here to be reflective of the field’s understanding of dystonia pathogenesis.

10) Lines 125-126: It is unclear to me what the authors mean by “the final common pathway” here. I assume that they are referring to some pathway that is shared by all dystonias? Is this known?

11) Line 127: What “failure” are the authors referring to in this sentence? It is unclear from the preceding text.

12) Line 128: The authors should include a period after “vs” here.

13) Line 131: What do the authors mean by “widely accepted description” here? I realize that they are talking about how dystonia is classified, but it is not made clear in the text. Also, what do the authors mean by “in the first instance” here? First instance of what?

14) Line 140: The statement, “TOR1A variations, due to ANO3 mutations” suggests that ANO3 mutations lead to the generation of TOR1A variations, which I do not believe is correct. In addition, all human genes need to be written in capital letters and in italics. This needs to be addressed throughout the manuscript.

15) Line 146: The statement, “other significant ones” here is strange, as it implies that Huntington’s disease is more significant than dystonia.

16) Lines 155-156: The sentence, “Dystonia usually develops on its own, but it can sometimes occur because of a stroke, an injury, or genetic factors is redundant with several earlier statements in this manuscript.

17) Line 160: What do the authors mean by “extended duration” here? How long is “extended”?

18) Line 162: In the section “3. The genetic background of dystonia”, the authors should briefly explain what is known about the functions performed by the proteins that they are discussing (e.g., torsinA, THAP1, etc.). 

19) Line 173: The statement, “have undergone extensive replication and validation” is strange, as it implies that the four previously mentioned genes have undergo an amplification via replication.

20) Line 178: The authors should replace “resulted” with “results” here.

21) Line 179: The authors should replace “in a preserved area” with “in the conserved C-terminus” here.

22) Line 181: The authors should provide some more information regarding the variants listed here.

23) Lines 183-184: The last sentence in this paragraph feels redundant with the second to last sentence.

24) Line 186: The authors need to replace “genes” here with “gene”.

25) Lines 191-192: The statement, “The primary craniocervical symptoms of THAP1” is strange, as it implies that the THAP1 gene has symptoms.

26) Line 198: The authors should replace “cervical, cranial, and larynx” with “cervical, cranial, and laryngeal regions”.

27) Line 204: The authors need to insert an “in” following “found” in this sentence.

28) Line 205: The authors statement, “All three mutations identified there” is confusing and needs more context. Mutations in GNAL?

29) Line 209: The authors should insert an “a” following “with” in this line.

30) Line 216: The authors should define the abbreviation “HOPS” used here.

31) Lines 222-224: Here, the authors state, “This fact confirms that genes are highly associated with the onset of focal dystonia…”. They should be a bit more specific regarding “genes” here.

32) Line 227: The authors should define the abbreviation “RIM” used here.

33) Lines 232-233: The authors’ statement, “which is known to impede neurotransmission” needs to be fleshed out a bit more. Where is neurotransmission defective exactly?

34) Lines 242-243: The statement, “a novel mutation in ATP1A3 was discovered” need to be further unpacked. Where was this mutation discovered?

35) Line 248: The authors’ statement, “supplies instructions for its synthesis” is unclear. How exactly does GCH1 instruct the production of BH4?

36) Line 252: When the authors state, “caused by BH4” here, do they mean “caused by a lack of BH4” or something else?

37) Line 259: The authors need to define “SGCE-M-D”, as this abbreviation has not yet been introduced in the manuscript.

38) Line 279: The authors should insert a “The” at the start of this sentence.

39) Lines 291-293: The authors need to better explain “Effect by voluntary action, overflow/mirror dystonia, and a “null point” are among the clinical characteristics of dystonia” for any non-experts who may want to read their review article. 

40) Line 322: What do the authors mean by “off-state dystonia” here?

41) Line 327: The authors need to change “have” to “has” to maintain subject-verb agreement in this sentence. In addition, the authors need to be consistent in their use of abbreviations. For example, the abbreviation BoNT introduced here should be used instead of botulinum neurotoxin in the remainder of the text.

42) Lines 330-331: The statement “cleavage of the soluble N-ethylmaleimide-sensitive factor attachment protein (SNAP) receptor complex’s components” need to be better unpacked by the authors. 

43) Line 332: A “The” needs to be inserted at the beginning of the sentence that starts “Main impact” and the “Main” needs to be written in lowercase letters.

44) Line 349: What is “immunerrorogenicity”?

45) Line 351: When the authors state, “frequency of antibody neutralization” are they referring to the neutralization of BoNT?

46) Line 352: When the authors state, “not all aspects of antibody formation are fully understood”, are they referring to the formation of anti-BoNT antibodies?

47) Line 354: The authors need to change “per injecting” to “per injection” here.

48) Line 357: The authors need to be consistent in their use of abbreviations. For example, the abbreviation DBS introduced here should be used instead of deep brain stimulation in the remainder of the text.

49) Line 363: The authors should replace “hasn’t” with “has not”.

50) Lines 365-366: The sentence “In this field, there are just two research with the strongest degree of evidence” is unclear and needs to be reworded.

51) Line 374: The authors should replace “5HT” with “serotonin” as they do not define “5HT” previously.

52) Line 377: How exactly does Dipraglurant “affect” mGluR5? The authors should be more specific.

53) Table 1:

a.    The placement of “Primary focal dystonia” should be changed such that it is found above the different primary focal dystonias rather than the treatments. 

b.    A title for the treatments/therapies is needed.

c.     For each primary dystonia, are certain therapies better or worse than others? Rather than simply using a “+/-“ criteria, perhaps the authors could include this information as increasing numbers of “+” signs?

54) Line 396: The word “indicated” here might be better if it were replaced with “suggested” or “recommended”.

55) Line 406: The authors should insert “to be” following “shown” in this sentence.

56) Line 410: The discussion of “bone grafting” feels like it comes out of nowhere here. Can the authors elaborate more on what they are trying to communicate here?

57) Lines 419-420: Why do the authors limit their consideration of therapies for dystonia here to ones caused by deletions? Would not point mutants also be addressable? Also, the authors should elaborate on what the aforementioned “nanoparticles or adenoviruses” might be carrying as cargo.

58) Line 421: What do the authors mean by “the most developed field of gene therapy” here?

59) Line 424: The authors need to briefly explain what “gendicine” is here.

60) Figure 2:

a.    What is “fixed dystonia”? This term needs to be defined in the text.

b.    The text box found at the bottom right of this figure is problematic, as the text in the box ends prematurely.

c.     I do not see the purpose of the drawing of the person in this figure.

61) Figure 3:

a.    The different Levels (i.e., A, B, C, and U) mention in this figure are not discussed anywhere in the text. The authors should address this issue, otherwise these levels feel like they come out of nowhere.

Comments on the Quality of English Language

The quality of English language needs a little bit of improvement, as indicated in my comments.

Author Response

Response to the Reviewer 1:

According to the Reviewer’s suggestions:

1)    My biggest criticism of this work is that the title indicates that the manuscript will focus on focal dystonias, while the main text of this manuscript feels more focused on dystonia in general. I think the authors should focus more on focal dystonias or change their title accordingly

We have changed the title as suggested by the Reviewer to: Genetic update and treatment for dystonia.

2)    Line 13: The term “Foundation genetic dystonias” is unclear to me. What do the authors mean?

The sentence was unclear. Thank you for bringing it to our attention, we have modified it as below:

“Genetic dystonias are linked to several genes, including pathogenic variations of VPS16, TOR1A, THAP1, GNAL, and ANO3.”

3)    Lines 14-15: When the authors state, “which can be challenging”, do they mean the “Diagnosis” or the “clinical signs”?

Thank you for pointing this out, we have changed the sentence to:

“Diagnosis of dystonia is primarily based on clinical symptoms, which can be challenging due to overlapping symptoms with other neurological conditions, such as Parkinson's disease.”

4)    Line 62: The authors should replace “Oromandibular dystonia (OMD)” with “OMD”, as they already defined “OMD” at the start of this paragraph.

We changed all the Oromandibular dystonia with OMD that we have found.

5)    Line 66: When the authors use “toxic”, do they mean “toxins”?

Yes, thank you very much for bringing this flaw by the Reviewer to our attention.

6)    Line 67: When the authors use “post-traumatic”, do they mean “post-traumatic stress disorder” or something else? 

Thank you, we were referring to post-traumatic stress disorder.

7)    Lines 109-111 and 118-119: The authors use of the term “spread” is confusing in these places. Are they meaning to say that the symptoms are spreading?

We are grateful to the Reviewer for pointing this out, of course, we meant to spread the symptoms. We have altered the sentences as follows: 

“Additionally, findings indicate that individuals with BSP tend to spread symptoms to the neck and oromandibular area […]”

8)    Lines 110-111: The authors might consider replacing “on the other hand” here with “alternatively”, as it reads a bit confusing considering the use of “hand” just prior.

We have changed the sentence following the Reviewer's advice. Indeed, the choice of words was unfortunate at the very least.

9)    Line 125: The authors should replace “control” with “controls” here to maintain subject-verb agreement. In addition, it seems problematic that the authors limit their discussion to the basal ganglia, as several other brain regions (not to mention the spinal cord) are known to be involved in dystonia pathogenesis. The authors should incorporate a more holistic view here to be reflective of the field’s understanding of dystonia pathogenesis.

We have expanded the information on the pathogenesis of dystonia as follows:

“Concepts of the neuroanatomical basis of dystonia have changed from a relatively limited focus on basal ganglia dysfunction to a more comprehensive motor network model in which the basal ganglia, cerebellum, cerebral cortex, and other brain regions play important roles. Our knowledge of the underlying physiological abnormalities has increased and several changes in brain signaling in the cortex, cerebellum, and basal ganglia have been identified [28].”

10) Lines 125-126: It is unclear to me what the authors mean by “the final common pathway” here. I assume that they are referring to some pathway that is shared by all dystonias? Is this known? and 11) Line 127: What “failure” are the authors referring to in this sentence? It is unclear from the preceding text.

We have removed the unclear sentences, thank you for bringing it to our attention.

12) Line 128: The authors should include a period after “vs” here.

We have added a punctuation mark, thank you.

13) Line 131: What do the authors mean by “widely accepted description” here? I realize that they are talking about how dystonia is classified, but it is not made clear in the text. Also, what do the authors mean by “in the first instance” here? First instance of what?

We have altered the sentence as follows:

“The use of the term 'primary dystonia' has become less common since the release of the more modern Albanese classification, which provides a different classification that might only differentiate between acquired and hereditary causes of dystonia [25,26].”

14) Line 140: The statement, “TOR1A variations, due to ANO3 mutations” suggests that ANO3 mutations lead to the generation of TOR1A variations, which I do not believe is correct. In addition, all human genes need to be written in capital letters and in italics. This needs to be addressed throughout the manuscript.

Thank you for pointing this out, indeed an unfortunate selection of words. Additionally, genes that we found have been corrected in italics.

15) Line 146: The statement, “other significant ones” here is strange, as it implies that Huntington’s disease is more significant than dystonia.

We have revised this sentence, thank you for drawing attention to this unfortunate statement.

16) Lines 155-156: The sentence, “Dystonia usually develops on its own, but it can sometimes occur because of a stroke, an injury, or genetic factors is redundant with several earlier statements in this manuscript.

We have removed the recurring sentence.

17) Line 160: What do the authors mean by “extended duration” here? How long is “extended”?

We have removed this information as it contributed nothing to the value of the section.

18) Line 162: In the section “3. The genetic background of dystonia”, the authors should briefly explain what is known about the functions performed by the proteins that they are discussing (e.g., torsinA, THAP1, etc.). 

We have added this information in parentheses next to the numeration.

19) Line 173: The statement, “have undergone extensive replication and validation” is strange, as it implies that the four previously mentioned genes have undergo an amplification via replication.

We have corrected the phrase as follows:

“Currently, genes THAP1, GNAL, TOR1A, and ANO3 are linked to isolated dystonia and have undergone extensive validation. KMT2B and VPS16 are also well validated, bringing the total number of monogenic dystonias to at least 6, plus a longer list of combined dys-tonias such as PANK2, ATP1A3, SCGE, etc.”

20) Line 178: The authors should replace “resulted” with “results” here. and 21) Line 179: The authors should replace “in a preserved area” with “in the conserved C-terminus” here.

We have changed the wording to that suggested by the Reviewer.

22) Line 181: The authors should provide some more information regarding the variants listed here.

We have added information about these variants, as below:

“A functional missense variant known as rs1801968 exists in exon 4 of the TOR1A gene, where histidine is substituted by aspartic acid at position 216. On the other hand, the precise functional effect of rs1182 on primary dystonia is still unknown, as there is no known functional effect of rs1182 on TOR1A expression and function  [41].”

23) Lines 183-184: The last sentence in this paragraph feels redundant with the second to last sentence.

We have removed a recurrent sentence.

24) Line 186: The authors need to replace “genes” here with “gene”.

Replaced, thank you.

25) Lines 191-192: The statement, “The primary craniocervical symptoms of THAP1” is strange, as it implies that the THAP1 gene has symptoms.

We have modified the sentence to: 

“Mutations of the THAP1 gene are characterized by juvenile-onset global dystonia, and the condition is mostly linked to missense variants in the gene [43]. “

 26) Line 198: The authors should replace “cervical, cranial, and larynx” with “cervical, cranial, and laryngeal regions”.

Replaced, thank you.

27) Line 204: The authors need to insert an “in” following “found” in this sentence.

Inserted, thank you.

28) Line 205: The authors statement, “All three mutations identified there” is confusing and needs more context. Mutations in GNAL?

Thank you for pointing out, we did not add the most important thing, which was the mutations.

“All three mutations identified there (c.677G>T (p.Cys226Phe), c.1315G>A (p.Val439Met) and c.1288G>A (p.Ala430Thr)) are thought to be pathogenic, and the clinical profile fits the characteristics already linked with GNAL variants [46]. “

30) Line 216: The authors should define the abbreviation “HOPS” used here.

We have expanded the abbreviation, thank you.

31) Lines 222-224: Here, the authors state, “This fact confirms that genes are highly associated with the onset of focal dystonia…”. They should be a bit more specific regarding “genes” here.

We have removed this information as it was our opinion that contributed no content value to the text.

32) Line 227: The authors should define the abbreviation “RIM” used here.

We have expanded the abbreviation, thank you.

33) Lines 232-233: The authors’ statement, “which is known to impede neurotransmission” needs to be fleshed out a bit more. Where is neurotransmission defective exactly?

We have added further clarification as follows:

“Complete deletion of RIMBP1 impedes neurotransmission via the link between presynaptic action potentials and the exocytosis of synaptic vesicles, as well as via presynaptic voltage gated Ca2+ channels (VGCCs). This results in lower numbers of cerebellar synapses decreased Purkinje cell dendritic arborization, and motor disorders resembling dystonia in mice [49].”

34) Lines 242-243: The statement, “a novel mutation in ATP1A3 was discovered” need to be further unpacked. Where was this mutation discovered?

We have added further explanation as follows:

“In the ATP1A3 A813V mutation, the initial alanine was changed to valine. Crystallography showed that the side chain of valine was significantly larger than that of alanine, which changed the shape of the ATP1A3 protein and reduced its biological activity [51].”

35) Line 248: The authors’ statement, “supplies instructions for its synthesis” is unclear. How exactly does GCH1 instruct the production of BH4?

We have tried to make the sentence more explicit, as follows:

“The GCH1 gene codes for the expression of the enzyme GTP cyclohydrolase 1. This enzyme is involved in the first of three steps in the formation of a compound known as tetrahydrobiopterin (BH4). For the synthesis of important neurotransmitters like dopamine, serotonin, and nitric oxide synthases, BH4 is a necessary component.”

36) Line 252: When the authors state, “caused by BH4” here, do they mean “caused by a lack of BH4” or something else?

 We were referring to those caused by the absence of BH4.

37) Line 259: The authors need to define “SGCE-M-D”, as this abbreviation has not yet been introduced in the manuscript.

We have expanded the abbreviation, thank you.

38) Line 279: The authors should insert a “The” at the start of this sentence.

Added, thank you.

39) Lines 291-293: The authors need to better explain “Effect by voluntary action, overflow/mirror dystonia, and a “null point” are among the clinical characteristics of dystonia” for any non-experts who may want to read their review article. 

We have unfolded the phrase, as follows:

“While imaging and laboratory tests are usually normal in patients with dystonia, there is no good diagnostic test for idiopathic dystonia. Voluntariness, mirror dystonia (unilateral abnormal posture induced by contralateral movements), overflow dystonia (involuntary contraction of muscles anatomically distinct from the primary movement), and, in certain cases, dystonic tremor (movements are oscillatory but not strictly rhythmic, jerky and patterned) are key clinical features of dystonia. In addition, there may be a "null point" where a person's movements are worse in one position and better in another due to dystonia or dystonic tremor. The presence of relaxation techniques, also known as sensory tricks is another essential component.”

40) Line 322: What do the authors mean by “off-state dystonia” here?

We expanded the phrase as follows:

“Of these, 30% generate off-state dystonia (which occurs when there are low levels of plasma levodopa, and respond well to levodopa treatment), and 10% to 15% may develop (usually lower) limb dystonia. “

41) Line 327: The authors need to change “have” to “has” to maintain subject-verb agreement in this sentence. In addition, the authors need to be consistent in their use of abbreviations. For example, the abbreviation BoNT introduced here should be used instead of botulinum neurotoxin in the remainder of the text.

We changed all the “botulinum neurotoxin” with abbreviations that we have found. 

42) Lines 330-331: The statement “cleavage of the soluble N-ethylmaleimide-sensitive factor attachment protein (SNAP) receptor complex’s components” need to be better unpacked by the authors. 

We changed sentence as follows:

“The mechanism of action of BoNT involves extracellular binding to glycoprotein structures on cholinergic nerve terminals, cleavage of components of the SNAP (soluble N-ethylmaleimide-sensitive factor attachment protein) receptor complex, and neuromuscular transmission by intracellular blockade of acetylcholine release.”

43) Line 332: A “The” needs to be inserted at the beginning of the sentence that starts “Main impact” and the “Main” needs to be written in lowercase letters.

Added, thank you.  

44) Line 349: What is “immunerrorogenicity”?

It was a misspelling typo; we have corrected this error.

45) Line 351: When the authors state, “frequency of antibody neutralization” are they referring to the neutralization of BoNT?

Yes, we added this information.

46) Line 352: When the authors state, “not all aspects of antibody formation are fully understood”, are they referring to the formation of anti-BoNT antibodies?

Yes, we added this information. 

47) Line 354: The authors need to change “per injecting” to “per injection” here.

 Changed, thank you.

48) Line 357: The authors need to be consistent in their use of abbreviations. For example, the abbreviation DBS introduced here should be used instead of deep brain stimulation in the remainder of the text.

We changed all the deep brain stimulation with abbreviations that we have found. 

49) Line 363: The authors should replace “hasn’t” with “has not”.

It was replaced, thank you.

50) Lines 365-366: The sentence “In this field, there are just two research with the strongest degree of evidence” is unclear and needs to be reworded.

At the suggestion of the second Reviewer, we have removed this sentence and expanded on the DBS issue. 

51) Line 374: The authors should replace “5HT” with “serotonin” as they do not define “5HT” previously.

It is replaced, thank you.

52) Line 377: How exactly does Dipraglurant “affect” mGluR5? The authors should be more specific.

We have attempted to expand on this thought as follows:

“It has been suggested that targeting the metabotropic glutamate receptor type 5, or mGlu5, may be a viable therapeutic strategy for the treatment of several neurodegenerative diseases. Novel allosteric modulators have been shown to reduce disease-related pathology and improve cognitive function in preclinical models [88]. Dipraglurant, which affects the mGluR5 by negative allosteric modulation, contributes to the development of dystonia […]”

53) Table 1:

  1. The placement of “Primary focal dystonia” should be changed such that it is found above the different primary focal dystonias rather than the treatments. 
  2. A title for the treatments/therapies is needed.
  3. For each primary dystonia, are certain therapies better or worse than others? Rather than simply using a “+/-“ criteria, perhaps the authors could include this information as increasing numbers of “+” signs?

Unfortunately, we were unable to split a cell in the table diagonally to include both captions, so we changed the title to “Treatment” and altered the scale to add an increased +, rather than +/-. 

54) Line 396: The word “indicated” here might be better if it were replaced with “suggested” or “recommended”.

It was replaced, as suggested.

55) Line 406: The authors should insert “to be” following “shown” in this sentence.

It was added, as suggested.

56) Line 410: The discussion of “bone grafting” feels like it comes out of nowhere here. Can the authors elaborate more on what they are trying to communicate here?

We have removed this additional information as it did not add value to the content of the article and could be confusing for the reader.

57) Lines 419-420: Why do the authors limit their consideration of therapies for dystonia here to ones caused by deletions? Would not point mutants also be addressable? Also, the authors should elaborate on what the aforementioned “nanoparticles or adenoviruses” might be carrying as cargo.

We have added details as below. Deletion was our abbreviated thought, we have corrected the misleading information. 

“There are no reports of attempts being made to introduce gene therapies for diagnosed mutations in identified genes. The use of nanoparticles or adenoviruses as a vector with appropriate genetic material could be a promising therapy in dystonia caused by amendments in genes.”

58) Line 421: What do the authors mean by “the most developed field of gene therapy” here?

Gene therapy products are most used in the treatment of genetic diseases or cancer, where the investment in oncology treatment is far greater due to the spread of cancer in our population. This is also supported by the fact of extensive clinical research, resulting in the introduction of genitacine in China. More information about this topic is in article by one of our authors (https://doi.org/10.3390/cells12242803)

59) Line 424: The authors need to briefly explain what “gendicine” is here.

We added explanation as below:

“Gentacine is used to treat patients with head and neck squamous cell carcinoma associated with TP53 gene mutations. It is produced by Shenzhen SiBiono GeneTech and consists of recombinant adenovirus (rAd-p53) modified to encode the p53 protein. [97]. “

60) Figure 2:

  1. What is “fixed dystonia”? This term needs to be defined in the text.
  2. The text box found at the bottom right of this figure is problematic, as the text in the box ends prematurely.
  3. I do not see the purpose of the drawing of the person in this figure.

We have added information about fixed dystonia in the text before the figure. We have expanded the text box found at the bottom right of this figure, this is our oversight. We have also removed the person from the figure as suggested.

61) Figure 3:

  1. The different Levels (i.e., A, B, C, and U) mention in this figure are not discussed anywhere in the text. The authors should address this issue, otherwise these levels feel like they come out of nowhere.

We have removed this information from the figure. Recommendations given by Subcommittee of the American Academy of Neurology for the use of the listed formulations are given in parentheses, as the classification in the text would not be consistent.

We are grateful to the Reviewer for the comments that have enriched our manuscript. We hope that the alterations meet the reviewer's expectations.

Reviewer 2 Report

Comments and Suggestions for Authors

The manuscript "Genetic update and treatment for focal dystonia" by Koptielow et al is well written and concerns an interesting topic. I have a few comments and suggestion for improvement.

1. Page 5 the authors highlight the overlaps of Parkisnons and dystonia. Should mention that dystonia frequently manifests mild Parkinsonian features with references eg Haggstrom et al 2017

2. Line 170 the authors state "there are 4 monogenic dystonia genes" and then list several more. I would rephrase this as KMT2B and VPS16 are well validated brining the total number of monogenic dystonias to at least 6, plus a longer list of combined dystonias eg PANK2, ATP1A3, SCGE etc

3. Page 8 in clinical phenomenology tremor should be mentioned along with some comments about the typical characteristics of tremor in dystonia eg asymmetric, jerky, task/position specific, 

4. Figure 2. Should avoid the use of the term Primary dystonia and follow the 2013 Consensus classification hence "idiopathic isolated dystonia"

5. Authors may wish to replace one of the figures with a figure illustrating the currently accepted 2103 Dual Axis dystonia classification system

6. The treatment section needs more information on GPi and STN DBS for focal dystonias, especially cervical, Meige, blepharospasm, where numerous studies, (admittedly a small minority with delayed start sham stimulation design) have demonstrated significant benefit including patients refractory to Botulinum toxin. Limiting the comments on effectiveness to the few DBS studies with Class 1 design, unfairly underplays the effectiveness of DBS in medically refractory focal dystonia. 

7. The author do not mention thalamotomy for focal hand dystonia which has an extensive literature mainly from Japanese centres, using traditional RF lesioning and more recently MRgFUS. If the paper is intended be a comprehensive overview of treatment options, thalamotomy literature should be discussed.

Comments on the Quality of English Language

English is fine, would benefit from minor editing

Author Response

Response to the Reviewer 2:

According to the Reviewer’s suggestions:

  1. Page 5 the authors highlight the overlaps of Parkisnons and dystonia. Should mention that dystonia frequently manifests mild Parkinsonian features with references eg Haggstrom et al 2017

We have added the information found in the mentioned publication as follows:

“Parkinson's disease and dystonia are inextricably related [30]. There is strong evidence for the importance of dopaminergic dysfunction in the pathogenesis of dystonia, suggesting a possible pathophysiological overlap with parkinsonism. This evidence comes from animal models, neuropathological, neurophysiological, neuroimaging, and clinical studies. Clinically, parkinsonian features are an important component of some combination dystonias, including dopa-responsive dystonia. Both inherited isolated dystonias and idiopathic dystonias may have mild Parkinsonian features. Idiopathic isolated dystonia also often causes postural, action, and resting tremors. In addition to being common in idiopathic isolated dystonia, postural, action, and rest tremor may mimic PD tremors and be the cause of "scans without evidence of dopaminergic deficit". The clinical spectrum of motor features in dystonia will be broadened by the identification and better clinical characterization of Parkinsonian features in idiopathic and hereditary isolated dystonia. This may prevent misdiagnosis and guide future treatment research [31,32].”

  1. Line 170 the authors state "there are 4 monogenic dystonia genes" and then list several more. I would rephrase this as KMT2B and VPS16 are well validated brining the total number of monogenic dystonias to at least 6, plus a longer list of combined dystonias eg PANK2, ATP1A3, SCGE etc

We have corrected the inconsistent information and changed the sentence:

“Currently, genes THAP1, GNAL, TOR1A, and ANO3 are linked to isolated dystonia and have undergone extensive validation. KMT2B and VPS16 are also well validated, bringing the total number of monogenic dystonias to at least 6, plus a longer list of combined dystonias such as PANK2, ATP1A3, SCGE, etc.”

  1. Page 8 in clinical phenomenology tremor should be mentioned along with some comments about the typical characteristics of tremor in dystonia eg asymmetric, jerky, task/position specific,

We have expanded the statement on clinical phenomenology as described below:

“While imaging and laboratory tests are usually normal in patients with dystonia, there is no good diagnostic test for idiopathic dystonia. Voluntariness, mirror dystonia (unilateral abnormal posture induced by contralateral movements), overflow dystonia (involuntary contraction of muscles anatomically distinct from the primary movement), and, in certain cases, dystonic tremor (movements are oscillatory but not strictly rhythmic, jerky and patterned) are key clinical features of dystonia. In addition, there may be a "null point" where a person's movements are worse in one position and better in another due to dystonia or dystonic tremor. The presence of relaxation techniques, also known as sensory tricks is another essential component.”

  1. Figure 2. Should avoid the use of the term Primary dystonia and follow the 2013 Consensus classification hence "idiopathic isolated dystonia"

We have corrected this weakness, thank you for bringing this to our attention.

  1. Authors may wish to replace one of the figures with a figure illustrating the currently accepted 2103 Dual Axis dystonia classification system

We have added a figure for the Dual Axis Dystonia classification system, this is now Figure number 2 (also found below)

  1. The treatment section needs more information on GPi and STN DBS for focal dystonias, especially cervical, Meige, blepharospasm, where numerous studies, (admittedly a small minority with delayed start sham stimulation design) have demonstrated significant benefit including patients refractory to Botulinum toxin. Limiting the comments on effectiveness to the few DBS studies with Class 1 design, unfairly underplays the effectiveness of DBS in medically refractory focal dystonia.

Thank you for drawing attention to this fact. We unnecessarily added information resulting in a negative characterization of DBS in the treatment of dystonia, so we have therefore corrected the text as suggested by the reviewer and added information on the efficacy of STN and GPi DBS as below:

“Hereditary or idiopathic segmental or generalized dystonia has been treated with bilateral GPi DBS. Patients have shown a significant improvement in motor function, disability, and activities of daily living in both hereditary and idiopathic dystonia. Overall, the statistics suggest a 65% reduction in symptoms, which on average is long-lasting [83]. The follow-up was performed for 14 patients who received stimulation in GPi, STN, or both and had cervical or generalized dystonia. The average duration of subsequent studies was almost ten years. Results confirm that STN-DBS and GPi-DBS have comparable long-term effects and are safe for up to 15 years in the treatment of dystonia. It has been demonstrated that STN is a feasible, safe, and effective target that can be utilized in place of GPi in the treatment of both generalized dystonia and adult-onset cervical dystonia [84].”

  1. The author do not mention thalamotomy for focal hand dystonia which has an extensive literature mainly from Japanese centres, using traditional RF lesioning and more recently MRgFUS. If the paper is intended be a comprehensive overview of treatment options, thalamotomy literature should be discussed.

We have filled this deficiency by adding the following content. Thank you for drawing attention to this fact.

“In 1969, Siegfried et al. reported vo-thalamotomy for writer's cramp and observed a significant improvement in patients. The effectiveness of vo-thalamotomy for treating FHD in musicians, hair stylists, watchmakers, and table tennis players has now been demonstrated in several publications. In patients with aberrant posterior vo neuronal activity, lesioning of the posterior vo resulted in an instantaneous reduction of dystonia symptoms [86]. MR-guided focused ultrasound (MRgFUS) thalamotomy, which allows intracranial focal lesioning without an incision, is a less invasive and successful approach to treating tremors. Similar to radiofrequency thalamotomy, MRgFUS thalamotomy causes thermal lesions and is expected to have effects on FHD comparable to those of radiofrequency Vo-thalamotomy [87].”

We are grateful to the Reviewer for the comments that have enriched our manuscript. We hope that the alterations meet the reviewer's expectations.

Round 2

Reviewer 2 Report

Comments and Suggestions for Authors

The authors have produced a revised and improved paper incorporating all the suggested changes